# Switching from Adalimumab Originator to Biosimilar: Clinical Experience in Patients with Hidradenitis Suppurativa

**DOI:** 10.3390/jcm11041007

**Published:** 2022-02-15

**Authors:** Trinidad Montero-Vilchez, Carlos Cuenca-Barrales, Andrea Rodriguez-Tejero, Antonio Martinez-Lopez, Salvador Arias-Santiago, Alejandro Molina-Leyva

**Affiliations:** 1Hidradenitis Suppurativa Clinic, Dermatology, Hospital Universitario Virgen de las Nieves, 18012 Granada, Spain; dramonterovilchez@gmail.com (T.M.-V.); carloscuenca1991@gmail.com (C.C.-B.); andrea_13_3217@hotmail.com (A.R.-T.); antoniomartinezlopez@aol.com (A.M.-L.); alejandromolinaleyva@gmail.com (A.M.-L.); 2Instituto de Investigación Biosanitaria Granada, 18012 Granada, Spain; 3Dermatology Department, Faculty of Medicine, University of Granada, 18011 Granada, Spain; 4European Hidradenitis Suppurativa Foundation (EHSF), 06847 Dessau-Roßlau, Germany

**Keywords:** adalimumab, biosimilar, hidradenitis suppurativa, switching

## Abstract

Adalimumab is currently the only biological medicine approved by the FDA for the treatment of hidradenitis suppurativa (HS). The breakout of biosimilar drugs made them more accessible due to their impact on pharmacoeconomics. However, packaging, formulation, or excipients are unique characteristics of each drug. In that way, switching from adalimumab originator to biosimilar and between biosimilars could have implications in the clinical practice. The objective of this study is to describe our clinical experience in switching from adalimumab originator to biosimilar and switching back again. A single-center retrospective cohort study was conducted that included seventeen patients with HS treated with adalimumab originator in the maintenance phase, and that achieved Hidradenitis Suppurativa Clinical Response (HiSCR), who were switched to adalimumab biosimilar for no medical reasons. The reason for the change was to improve pharmacoeconomic efficiency, following our hospital policies on biologics. Median duration with adalimumab originator treatment before switching was 48 weeks. After switching, 41.2% of patients maintained HiSCR response without additional issues, while 58.8% (10/17) reported problems after the change. Switching from adalimumab originator to biosimilar in well-controlled patients could imply problems in efficacy and adherence. Switching back to adalimumab originator appears to solve most of the problems, but some patients can lose confidence in the drug and discontinue it. It would be worthwhile to evaluate the benefit–risk ratio individually when switching an HS patient to adalimumab biosimilar.

## 1. Introduction

Hidradenitis suppurativa (HS) is a recurrent, chronic, debilitating, inflammatory skin disease of the hair follicle that typically presents after puberty with deep-seated, painful, inflamed lesions in the apocrine gland-bearing areas of the body, most commonly the inguinal, axillae, and anogenital regions [1]. Adalimumab, a monoclonal antibody against tumor necrosis factor-α, is the only biologic agent currently available for treating moderate to severe HS approved by the FDA [2]. Adalimumab is also employed in other autoimmune diseases, such as psoriasis, rheumatoid arthritis, or inflammatory bowel disease [3].

Biologics have revolutionized the treatment of these diseases, but they are expensive drugs reserved for severe refractory patients [4]. Nevertheless, the breakout of biosimilar drugs, as they reach patent expiry, made biological drugs more available due to their impact on pharmacoeconomics, decreasing their impact on health care budgets [5,6]. A biosimilar drug is different from a bioidentical drug, meaning having the same molecular structure as a substance produced in the body.

Biosimilars are products highly similar in quality, safety, and efficacy to an already licensed biotherapeutic product [7]. Their manufacturing process includes complex preclinical studies to confirm the purity of the protein to be injected in humans. Nevertheless, it is only necessary to demonstrate the non-inferiority and safety of the drug in one licensed indication of the originator product, rheumatoid arthritis being the most used [8]. These different diseases for which they are indicated share an elevation of TNF-alpha in the inflammatory cascades, but they are different conditions, and the approved drug might not perform exactly equally in all these diseases, likely having implications on their safety and efficacy [9]. Moreover, biosimilar drugs are not an exact copy of the originator drug due to the heterogeneity of the production process and variations in manufacturing changes throughout their life cycles [8]. These slight particularities in their structure could have similar, noninferior, or even better action than the originator drug [10]. Although it has been proven in psoriasis patients that non-medical switch from adalimumab originator to adalimumab biosimilars was not associated with drug retention [11], the formulation, excipients, or packaging are unique characteristics of each drug, with potential implications for the patient use experience [12]. Because of this, switching from adalimumab originator to biosimilar and between biosimilars could have implications in clinical practice.

Clinical trials and published evidence from actual clinical practice suggest that post non-medical switch, treatment discontinuation rates for adalimumab are variable (between 6.1% to 55.9%) [13,14,15,16,17]. In addition, there is little evidence collected on patients returning to their original treatment after switching to a biosimilar (switch back). The main reasons for switching back are pain at the injection site, adverse reactions, and loss of efficacy [15]. The reported range of patients who switched back after non-medical switch with adalimumab was between 4% to 8.7% [17,18,19]. It is, therefore, necessary to describe the clinical experience regarding effectiveness, tolerability, and safety in routine clinical practice as in switching from originator adalimumab to biosimilar adalimumab in the different indications.

To our knowledge, there are no studies on the experience of non-medical switch and switch back in patients with HS that analyze tolerability and efficacy after switching. Therefore, the objective of this study is to describe our clinical experience in switching from adalimumab originator to biosimilar and switching back.

## 2. Materials and Methods

### 2.1. Study Design and Participants

A single-center retrospective observational study was designed between September 2019 and December 2020. It included all adult patients diagnosed with HS, on treatment with adalimumab originator, who were switched for non-medical reasons to adalimumab biosimilar in our HS clinic. The reason for the change was to improve pharmacoeconomic efficiency, following our hospital policies on biologics. Inclusion criteria: Patients with HS over 18 years old, receiving originator adalimumab in the maintenance phase (>12 weeks of treatment) with Hidradenitis Suppurativa Clinical Response (HiSCR) [2] who were switched to adalimumab biosimilar. Exclusion criteria: Patients with HS who did not sign the written consent form, receiving originator for less than 12 weeks, not achieving HiSCR response with adalimumab originator. All patients agreed with the treatment regimen and signed a written consent form to use their personal data for the present study. This study was approved by the Ethics Committee of the Hospital Universitario Virgen de las Nieves (0139-N-21) and is in accordance with the Declaration of Helsinki.

### 2.2. Variables of Interest

Effectiveness, tolerability, and safety data are described. Clinical, sociodemographic, and biometric variables were recorded by means of clinical interview, physical examination, and cutaneous ultrasonography using a 7–15 MHz linear probe (myLab25 Esaote, Genova, Italy). Sociodemographic characteristics included sex, age, and smoking habit. Clinical characteristics included family history of HS, Hurley stage, disease duration, number of affected areas, nodules, abscesses and draining tunnels count, number of previous treatments. Effectiveness, tolerability, and safety data after switching to adalimumab biosimilar and after switching back to adalimumab originator are described. Changes between different types of adalimumab were made while maintaining the dosing regimen described in the datasheet [3], without performing a new induction period.

### 2.3. Follow-Up

Patients were evaluated every 12 weeks after switching from originator adalimumab to biosimilar, but if they had problems with the new drug, they could arrange non-scheduled visits in the HS Unit to be assessed by a dermatologist. Patients with effectiveness or tolerability issues after switching were offered to switch back to adalimumab originator. Following switching back, patients were reassessed after 12 weeks.

### 2.4. Statistical Analyses

Descriptive statistics were used to evaluate the sample characteristics. The Kolmogorov–Smirnov test was used to check the normality of the variables. Continuous data were expressed as the median (interquartile range) and qualitative variables as relative (absolute) frequencies. Student’s *t*-test or the Wilcoxon–Mann–Whitney test were employed to compare nominal and continuous data. The χ^2^ test or Fisher’s exact test, when necessary, were used to compare nominal data. Significance was established for all tests at two tails, *p* < 0.05. Statistical analyses were performed with JMP version 14.1.0 (SAS Institute, Cary, NC, USA).

## 3. Results

Twenty-two patients were evaluated for inclusion in the study, of which 77.3% (17/22) met the inclusion criteria. Of these, 70.6% (12/17) were male and 29.4% (5/17) female. Median age was 31 (19–51) years. Median duration with adalimumab originator treatment before switching was 48 (28–80) weeks. Clinical and sociodemographic characteristics are shown in Table 1. In relation to concomitant treatment, only one female patient had additional treatment with metformin 850 mg/24 h and spironolactone 50 mg/24 h. This patient maintained the HiSCr response after the switch. The remaining patients received adalimumab in monotherapy.

After switching, 41.2% (7/17) patients maintained HiSCR response without additional issues, while 58.8% (10/17) reported problems with the new drug. There were no differences in sociodemographic characteristics between both groups, Table 1.

Regarding switching problems, 23.5% (4/17) of patients had severe pain at the injection site, 23.5% showed loss of HiSCR response, 5.9% (1/17) had pain and loss of response simultaneously, and 5.9% (1/17) reported dizziness and nausea. The median time on biosimilar adalimumab in patients with tolerability or efficacy issues was 13.6 weeks before switching to originator.

In those patients who had tolerability or efficacy problems, it was proposed to switch back to adalimumab originator. Of the 10 patients with problems after the change, 80.0% (8/10) returned to their original treatment and were re-evaluated at 12 weeks and 20.0% (2/10) decided to discontinue treatment (one patient with loss of effect and one patient with pain and loss of effect) because they lost confidence in the drug. Patients switching flow chart is shown in Figure 1.

Of the eight patients who switched back, pain vanished in 100.0% of the cases (4/4) and 66.7% (2/3) regained HiSCR response. Only one patient did not recover response after 12 weeks of treatment, which was discontinued. In the patient who had dizziness and nausea, symptoms remitted after switching back.

## 4. Discussion

This study shows that some patients with well-controlled HS switching from adalimumab originator to adalimumab biosimilar could have effectiveness and tolerability problems. Pain at the injection site was the most frequent side effect, followed by lack of response. Most patients recovered from side effects after switching back, but 1/5 discontinued treatment after suffering side effects.

There is a lack of scientific evidence regarding switching in clinical practice. Regulatory agencies do not usually require switching studies to approve a biosimilar, so the effects of switching patients between biosimilars are unknown [18]. There are clinical trials that observed similar effectiveness and tolerability between originator adalimumab and biosimilar adalimumab in psoriasis [20]. However, the external validity of clinical trials might be low, as they usually have straight eligibility criteria, and the tool used to assess effectiveness may not be reliable in HS [21]. For example, clinical trials are frequently conducted in naïve patients to biologic treatment, and pregnant women, elderly, and very severe patients are not allowed to be included [20]. Moreover, it should be considered that switching data are not transferable between different biosimilars or diseases. In fact, large clinical trials do not have enough power to show differences in rare side effects [22].

To our knowledge, there are only two case series regarding biosimilar adalimumab use in HS patients. Ricceri et al. evaluated seven HS patients switching adalimumab originator to biosimilar SB5 adalimumab and found similar effectiveness rates compared to naïve HS patients treated with this biosimilar. Nevertheless, tolerability was not evaluated in this report [23]. Patil also reported effectiveness in two HS patients being treated with biosimilar ZRC-3197 adalimumab. However, they used adalimumab combined with methotrexate or doxycycline, which might have influenced the therapeutic effect. Tolerability was also not assessed [24]. Regarding switching with other biologic agents in HS patients, it has been recently reported that infliximab administration and infliximab-abda administration have similar safety and effectiveness data by Hidradenitis Suppurativa Clinical Response [25].

The rate of switching failure in our study was similar to that reported by Bergman et al. [14] and higher than in other reports on psoriasis, rheumatoid arthritis, and Crohn’s disease [13,15,16,17,26]. Differences in these rates could be due to the different diseases included [14]. Pain at the injection site was the most frequent side effect reported. Although it has been reported that it is related to citrate-containing adalimumab [14], patient sensitivity is also important. Pain greatly decreases patient quality of life [27], so the use of painless therapies is mandatory to succeed in patient management. Loss of response was the second most frequent side effect and 1/3 did not recover after switching back to adalimumab originator. Apart from the lack of a side effect recovery rate after switching back, it is also important to mention that two out of five patients decided to discontinue treatment due to side effects. Switching may alter patients’ confidence in the drug, causing them to decide to stop taking it. This is a fact of great importance in HS as there are currently no other approved biologics.

When deciding to switch biosimilar, it is also important to consider patients’ ability to adapt to changes [18]. In this study, no sociodemographic or clinical characteristics related to the lack of effectiveness of switching were observed. Currently, there is no study on this topic, so further investigation is needed to evaluate if demographic or clinical characteristics could be related to different adalimumab effectiveness and tolerability.

There are no studies regarding switching back in HS patients in clinical practice. Despite the risk of treatment discontinuation after switching, we observed that most patients recovered from side effects related to adalimumab biosimilar after switching back.

## 5. Conclusions

In conclusion, based on our results, switching from adalimumab originator to adalimumab biosimilar in patients with well-controlled HS could imply problems in adherence due to pain at the injection site or with the maintenance of clinical response. Switching back to adalimumab originator appears to solve most of the problems, but some patients can lose confidence in the drug and discontinue it. It would be worthwhile to individually assess the benefit–risk ratio of switching a well-controlled patient with hidradenitis suppurativa to adalimumab biosimilar. There is a need for more epidemiologic studies in this field.

## Figures and Tables

**Figure 1 jcm-11-01007-f001:**
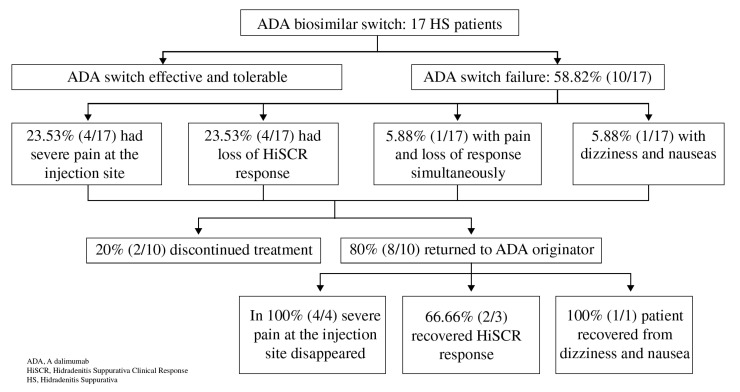
Patients switching flow chart.

**Table 1 jcm-11-01007-t001:** Sociodemographic and clinical characteristics of the patients.

Variables	Total Sample (*n* = 17)	Switch Effective and Tolerable (*n* =7)	Switch Failure (*n* = 10)	*p*
Age (years)	31 (19–51)	43 (17–50)	26.5 (19–53.5)	0.675
Sex				
-Male -Female	12 (70.59%) 5 (29.41%)	5 (71.43%) 2 (28.57%)	7 (70%) 3 (30%)	1
Smoking habit (yes)	8 (47.06%)	2 (28.57%)	6 (60%)	0.335
Age of onset (years)	15 (15–22.5)	16 (15–33)	15 (14.25–18)	0.085
Family history (yes)	8 (47.06%)	4 (57.14%)	4 (40%)	0.637
Hurley stage				
-I -II -III	1 (5.88%) 4 (23.53%) 12 (70.59%)	1 (14.29%) 1 (14.29%) 5 (71.43%)	0 3 (30%) 7 (70%)	0.394
AN count	2 (0.5–6.5)	2 (0–9)	3 (0.75–5.75)	0.588
Draining tunnels count	3 (2–4.5)	3 (1–9)	2.5 (2–3.25)	0.129
Number of affected areas	4 (3–4)	4 (2–4)	4 (3.75–4.25)	0.473
Number of previous treatments	4 (2.5–4.5)	4 (3–5)	4 (2–4.25)	0.429
Follow-up time before switching (weeks)	48 (28–80)	32 (20–80)	48 (43–87)	0.167

AN, total abscess and inflammatory nodule count. Data are expressed as relative (absolute) frequencies and median (interquartile range). Student’s *t*-test for independent samples or the Wilcoxon test were used to compare continuous variables, depending on the normality of the variable. The chi-square test or Fisher’s exact test, as appropriate, were applied to compare categorical data. A two-tailed *p* < 0.05 was considered statistically significant for all tests.

## Data Availability

Data could be available by request to the corresponding author.

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
