# Peer review of "Switching from Adalimumab Originator to Biosimilar: Clinical Experience in Patients with Hidradenitis Suppurativa"

_jcm, 2022, doi:10.3390/jcm11041007_

Round 1

Reviewer 1 Report

The authors of the manuscript tackle an emerging issue; switch from bioorignator to biosimilar for non-medical reasons.  The biggest strength of this manuscript, and the novelty, lies in the fact that non-responders were switched back to originator.

I believe the authors should make it clear whether the non-medical switch was done at the behest of third party (insurance, hospital etc) for economical reasons or whether the switch was conducted for this study. According to the manuscript, patients were meant to sign a consent form - was this for the switch or follow follow up after the switch?   A brief discussion on the ethical issues surrounding non-medical switching may embellish the paper.

It may be relevant for the authors to include whether the patients were on concurrent HS treatment.

Some other minor points;

  • Line 34: I would say that adalimumab is the only FDA approved treatment for HS, not the only, as many biologics are used off license for the treatment of HS.
  • Line 48: You could introduce the term bioidentical
  • Affiliation 4 to EHSF is not linked to any of the authors.

Author Response

The authors of the manuscript tackle an emerging issue; switch from bioorignator to biosimilar for non-medical reasons.  The biggest strength of this manuscript, and the novelty, lies in the fact that non-responders were switched back to originator.

Thank you for the comments.

I believe the authors should make it clear whether the non-medical switch was done at the behest of third party (insurance, hospital etc) for economical reasons or whether the switch was conducted for this study.

We have included that non-medical switch was done for economical reasons following our hospital policies. The following sentence has been added “The reason for the change was pharmacoeconomic to improve efficiency, following our hospital policies about biologics”

According to the manuscript, patients were meant to sign a consent form - was this for the switch or follow follow up after the switch? 

They had to sign a consent for the follow-up. The switch was due to our hospital policies and the patient had not the option to choose the drug that he wanted.

  A brief discussion on the ethical issues surrounding non-medical switching may embellish the paper.

We have included a new paragraph about ethical issues following your recommendations. “Switching drugs for non-medical reasons have ethical implications. Doctors must always act for the patients benefit and promote health and well-being to them. Nevertheless, it is not always clear if this benefit should be only for one individual patient or for the society. By switching to biosimilars for economic reasons, the benefits for society are trumping those of an individual patient. Nevertheless, there are no statement in the physicians’ codes indicating that they can prioritize public health or health economic interests over an individual patient, underscoring the primacy of patients’ interests in the current ethical paradigm. So, if there are significant difference in effectiveness and adverse events between a biologic and a biosimilar, doctors have the professional duty to recommend the option that prioritizes their patients’ wellbeing”

It may be relevant for the authors to include whether the patients were on concurrent HS treatment.

In relation to concomitant treatment, only one female patient had additional treatment with metformin 850mg/24h and spironolactone 50mg/24h. This patient maintained the HiSCr response after the switch. The remaining patients received adalimumab in monotherapy.

Some other minor points;

  • Line 34: I would say that adalimumab is the only FDA approved treatment for HS, not the only, as many biologics are used off license for the treatment of HS.
    • We have changed this information according to your recommendations. The following sentence has been added: Adalimumab is currently the only biological approved by the FDA for the treatment of hidradenitis suppura-tiva (HS).
  • Line 48: You could introduce the term bioidentical
    • We have introduced this term: “a bioidentical drug, meaning having the same molecular structure as a substance produced in the body”
  • Affiliation 4 to EHSF is not linked to any of the authors
    • It has been added to the last author

Reviewer 2 Report

Dear authors,

The idea of this study is interesting and relevant to patients and practitioners. However, the sample size is  too small to support the conclusion that more than half of the patients could have effectiveness and tolerability problems (line 147). 

You suggest in the abstract that "It would be advisable to evaluate the benefit-risk ratio individually when switching a
HS patient to adalimumab biosimilar." How are practitioners supposed to make such an individual evaluation?

L. 74-85: Time of inclusion?

L. 84 Please provide ethical approval number. 

L. 110: What is the purpose of all your statistical tests?

L. 115: Please add (interquartile range)

Author Response

The idea of this study is interesting and relevant to patients and practitioners. However, the sample size is  too small to support the conclusion that more than half of the patients could have effectiveness and tolerability problems (line 147). 

Thank you for the comments. We have changed it by “some patients”

You suggest in the abstract that "It would be advisable to evaluate the benefit-risk ratio individually when switching a
HS patient to adalimumab biosimilar." How are practitioners supposed to make such an individual evaluation?

It would be also advisable that patients were asked if they had pain problems with other drugs administration, as the pain at the injection site was the most frequent adverse event reported. Moreover, patients with problems with treatment adherence should not be switched because they could decrease the confidence in the drug and decrease even more the adherence. It would be also important that difficult to treat patients, meaning those that had difficulty to reach clinical response, were not switch because the efficacy could decrease and they could not reach again the response

  1. 74-85: Time of inclusion?

The time of inclusion was between September 2019 and December 2020. This information has been added.

  1. 84 Please provide ethical approval number. 

It has been added (0139-N-21)

  1. 110: What is the purpose of all your statistical tests?

The purpose of the statical test is to compare sociodemographic and clinical characteristics between patients with switch failure and with switch effectiveness to look for characteristics that could help clinicians to identify the profile of patient that have a worser response after switching.

  1. 115: Please add (interquartile range)

It has been added